# Synthesis and Properties of Injectable Hydrogel for Tissue Filling

**DOI:** 10.3390/pharmaceutics16030430

**Published:** 2024-03-21

**Authors:** Chunyu Xie, Ga Liu, Lingshuang Wang, Qiang Yang, Fuying Liao, Xiao Yang, Bo Xiao, Lian Duan

**Affiliations:** State Key Laboratory of Resource Insects, College of Sericulture, Textile and Biomass Sciences, Southwest University, Chongqing 400715, Chinalfy12314x@email.swu.edu.cn (F.L.); yx198329@swu.edu.cn (X.Y.)

**Keywords:** hydrogel, injectability, tissue filling

## Abstract

Hydrogels with injectability have emerged as the focal point in tissue filling, owing to their unique properties, such as minimal adverse effects, faster recovery, good results, and negligible disruption to daily activities. These hydrogels could attain their injectability through chemical covalent crosslinking, physical crosslinking, or biological crosslinking. These reactions allow for the formation of reversible bonds or delayed gelatinization, ensuring a minimally invasive approach for tissue filling. Injectable hydrogels facilitate tissue augmentation and tissue regeneration by offering slow degradation, mechanical support, and the modulation of biological functions in host cells. This review summarizes the recent advancements in synthetic strategies for injectable hydrogels and introduces their application in tissue filling. Ultimately, we discuss the prospects and prevailing challenges in developing optimal injectable hydrogels for tissue augmentation, aiming to chart a course for future investigations.

## 1. Introduction

In recent years, research about skin rejuvenation, such as restoring tissue volume, diminishing wrinkles, and addressing skin laxity, has become a hot topic due to growing demand [1,2]. Restoring tissue defects or altering the overall appearance can be improved through surgical treatments or minimally invasive procedures. Traditional surgical approaches like filler implantations necessitate surgical incisions and are frequently associated with complications, including postoperative infections, prolonged recovery periods, and peripheral nerve damage [3,4,5]. In contrast, minimally invasive techniques based on injectable fillers have become popular for treating early aging and reversing tissue volume loss [6], owing to the minimal adverse effects, faster recovery, good results, and negligible disruption to daily activities [7,8,9,10].

Injectable fillers are diverse and include hydrogels [11], autologous fats [12], autologous platelet-rich plasma [13], and microspheres [14] and are particularly noteworthy due to their unique advantages. Hydrogels are composed of a polymer network and immobile water inside. The higher water content offers an ideal environment for cell survival, and their unique polymer network possesses regulable elasticity to emulate the mechanical properties of the natural extracellular matrix (ECM), offering essential support to the surrounding cells [15,16]. Injectable hydrogels intended for soft-tissue augmentation must exhibit analogous mechanical properties to those of the target tissues and degrade slowly to maintain volume in vivo. Rather than merely occupying space, these hydrogels interact with host cells to perform functions, like promoting subcutaneous cell proliferation, collagen deposition, and tissue regeneration [17,18].

Nowadays, injectable hydrogels based on different types of polymers are ubiquitous in the cosmetic and medical fields. These injectable hydrogels are not only applied for esthetic surgeries but also for tissue engineering, including congenital defects and trauma and surgical removal, while focusing on the repair of damaged tissues. Despite significant progress, injectable hydrogels still require more research for optimizing their therapeutic potential. This review aims to offer a brief summary of injectable hydrogels as tissue fillers. The corresponding mechanisms, classifications, physicochemical properties, and biological functions of these hydrogels are also discussed.

## 2. Synthesis Mechanisms of Injectable Hydrogels

In the past, the administration of hydrogels for tissue filling heavily depended on invasive surgical procedures, which may lead to serious postoperative complications due to extensive incisions [3,19]. Recently, injectable hydrogels with unique and special sol–gel transition ability or delayed gelatinization have been developed, offering a non-invasive approach for transferring hydrogels to the intended site. Injectable hydrogels are constructed through the interactions between polymer chains. The nature of these interactions grants injectable hydrogels distinct advantages and specific limitations. For example, hydrogels based on physical interactions possess excellent biosafety and tissue adhesion, yet their undesirable mechanical strength may restrict their filling effect [20]. Conversely, chemically crosslinked hydrogels exhibit enhanced stability and adjustable mechanical properties to match the ECM stiffness, but the incorporation of some specific chemical bonds may increase the cytotoxic risks [21]. Recently, hydrogels prepared by enzyme-induced crosslinking have attracted attention due to their innovative attributes. These enzyme-catalyzed processes enable the formation of covalent bonds that are green and biocompatible, providing researchers with a new avenue for developing advanced injectable hydrogels [22,23].

Particularly, the crosslinking ratio significantly influences the stability and mechanical properties of hydrogels. A higher crosslinking ratio in hydrogels results in a more compact structure with reduced swelling and increased mechanical strength. Therefore, the stability, mechanical strength, biodegradability, and bioactivity of hydrogels can be modulated by their crosslinking patterns [24,25]. The crosslinking density can also be varied via multiple crosslinking mechanisms, thereby enabling precise control over the hydrogel’s stiffness and softness [26]. The variation in these properties often dictates their specific applications in biomedical fields.

### 2.1. Chemical Covalent Crosslinking

Chemical covalent crosslinking enables the production of stable hydrogels, facilitating the physical matching with different types of tissues. The injectable character can be achieved via some dynamic covalent chemical bonds, which endow the hydrogel with a reversible sol–gel transformation to pass through the syringe needle and gelatinize in situ. Additionally, liquid precursors for radical polymerization or delayed gelatinization can also smoothly pass through the syringe, subsequently generating a hydrogel in situ after the polymerization or crosslinking.

#### 2.1.1. Dynamic Covalent Chemical Bonds

Dynamic covalent chemical bonding, like the Michael addition, Diels–Alder (DA) reaction, and Schiff base reactions, is commonly utilized to develop injectable hydrogels. The Michael addition, a nucleophilic addition reaction based on Michael receptors, does not require chemical catalysts or initiators with potential cytotoxicity, which endows the hydrogels with good biocompatibility. Furthermore, the Michael addition reacts quickly under physiological conditions without generating by-products. The Michael addition facilitates the reversible crosslinking that is essential for hydrogel functionality through mechanisms like the thiol–disulfide exchange, where thiolate groups attack disulfide bonds, generating new disulfide bonds and thiolate groups [27]. Considering a limited number of terminal vinyl groups on linear polymers, hyperbranched and grafted macromolecules with abundant terminal groups are more frequently synthesized for the fabrication of injectable hydrogels [28]. The Michael addition can be used to construct injectable protein or polypeptide hydrogels that mimic the mechanical strength of the ECM. For example, Yang et al. prepared 2-mercaptolethylamine hydrochloride-modified hyaluronic acid (HA) and dopamine-grafted HA [29]. The gelatinization occurred through Michael addition and electrostatic interactions when the two types of modified HA solutions were combined with poly (hexamethylene guanidine). Significantly, this hydrogel transitions to a liquid state under strains exceeding 520%, ensuring its injectability. After the injection, the destructed dynamic bonds based on Michael addition reactions gradually recovered, resulting in the re-established hydrogel. Similarly, Fu et al. utilized dynamic boronate ester linkages to construct composite hydrogel based on the Michael addition reaction (Figure 1A) [30]. This design enabled the hydrogel to transition from gel to sol under shear stress, as dynamic bonds were broken. Upon the removal of large external forces, these dynamic bonds quickly reformed to generate the hydrogel, thus achieving injectability.

Schiff base reactions facilitate the creation of dynamic bonding between nucleophilic groups, such as amine or hydrazine, and the electrophilic carbon atoms found in aldehydes or ketones [31]. This mechanism enables the hydrogel to undergo rapid and reversible sol–gel transformations by breaking down and subsequently restoring the crosslinked networks [32]. Under physiological conditions, Schiff base linkages between the aldehyde groups and amines formed rapidly, which provides a simple and reliable method for the formation of cell-friendly materials [33,34]. Wei et al. demonstrated that the amino group on *N*-carboxyethyl chitosan and aldehyde group on oxidized sodium alginate formed reversible bonds through the Schiff base reaction, thereby constructing injectable polysaccharide hydrogels [35]. During injection, these dynamic bonds broke, allowing for easy passage through the syringe. After injection, the bonds reversibly reformed and gelatinized in situ. Zhang et al. modified poly (ethylene glycol) (PEG) with 4-formylbenzoate and prepared an agarose–ethylenediamine conjugate (AEC) [36]. Mixing this modified PEG with AEC led to dynamic Schiff-based crosslinking between the aldehyde group of the modified PEG and the amine groups on AEC, resulting in hydrogel formation. During the injection process, Schiff base crosslinking dissociated under shear stress, which ensured that the hydrogel passed through a 20G needle. After injection, the disrupted hydrogel recovered to the integrated hydrogel through regenerated Schiff base crosslinking. The self-healing property endowed the hydrogel with sufficient mechanical strength to support surrounding tissues after the injection.

The DA reaction, a reversible click chemistry approach involving a conjugated diene and dienophile, typically an alkene or alkyne, is also utilized in the synthesis of injectable hydrogels [37,38]. This reaction is widely employed due to its high yield, harmless secondary products, simple reaction conditions, availability of raw materials and reaction reagents, and fast synthesis reaction [39]. Notably, the DA reaction proceeds under mild conditions without any catalyst or coupling agents, which is considered a kind of biocompatibility reaction and can realize substrate binding to specific biomolecules [40]. For example, Ghanian and colleagues adopted furan-modified alginate and maleimide-grafted PEG to initiate the DA click reaction [41]. After mixing the modified sodium alginate and PEG aqueous solution with Ca^2+^ ions, ionic crosslinking started rapidly, and the DA reaction gradually induced gelation, endowing the mixture with shear-thinning and self-healing ability. The ionic interactions were sacrificed during injection, after which the DA reaction between furan and maleimide autonomously occurred to regenerate hydrogel under physiological conditions.

Dynamic chemical bonds, such as acylhydrazone and boronate bonds, are also utilized in the synthesis of injectable hydrogels due to their reversible nature, which is responsive to environmental stimuli. Acylhydrazone bonds, formed between hydrazine group and aldehyde or the ketone group, exhibit dynamic covalent bonding with temperature and pH response characteristics [42,43]. Jiang and co-workers prepared cellulose-based hydrogels with injectable ability through reversible acylhydrazone bonding [44]. After injection, the dynamic hydrazone bonds quickly self-healed and gelatinized in situ. Boronate bonds are reversible covalent bonds based on diols and boric acids [37]. Sun et al. created dynamic covalently bonded hydrogels by mixing chitosan, alginate, and formylphenylboronic acid under mild conditions (Figure 1B) [45]. Owing to the imine borate and boronic ester bonding, the hydrogel transitioned between the sol and gel state before and after injection, resulting in excellent injectability.

#### 2.1.2. Radical Polymerization

During radical polymerization, free radicals initiate the process by extracting electrons from the double bonds of monomers. This action not only bonds neighboring monomers together but also generates new radicals at the opposite end of the involved bonding. The new radical continues to attack similar unsaturated groups, resulting in the polymerization of monomers and the formation of hydrogel. For injectable hydrogel, the precursor monomer solution with photo-initiators can be injected in the liquid state [46]. After injection, the photo-initiators generate radicals with the assistance of ultraviolet or visible light, inducing polymerization and the formation of hydrogel in situ [47]. Noshadi and co-workers mixed gelatin methacryloyl and Eosin Y (photoinitiator) for injection [48]. After injection, the mixture was photopolymerized in situ with visible light, resulting in the formation of a hydrogel (Figure 1C). Similarly, Liu et al. prepared methacrylate sodium alginate (AMSA) by grafting methacrylate groups [49]. After injection, the methacrylate groups in AMSA polymerized under ultraviolet light, leading to the formation of hydrogel. The abundant free radicals in the system are crucial for the formation of hydrogels based on the radical polymerization. However, the application of these hydrogels in tissue filling is restricted due to the limited degradation capacity and unavoidable toxicity resulting from initiators.

#### 2.1.3. Delayed Gelatinization

Some injectable hydrogels possess the intrinsic characteristic of being a “smart” delivery vehicle, stemming from their responsiveness to external or internal stimuli [50]. Polymers with stimuli responsiveness have the ability to undergo sol–gel transformations in response to changes in external physicochemical parameters, such as temperature or pH value [51]. Therefore, they can exist as a liquid precursor for injection and subsequently undergo chemical crosslinking to transform into hydrogels in situ with the triggered stimuli. Similarly, precursors characterized by slower kinetics can also be administered in a flowing state and subsequently slowly generate chemical crosslinking to generate hydrogel with prolonged time. Stimulus-responsive hydrogels are typically fluid during injection, allowing for easy administration, and solidify into a stable hydrogel in situ after injection. Deng and colleagues developed an injectable hydrogel using HA and carboxymethyl cellulose (CMC) by introducing a thiolated natural polysaccharide ether (Figure 1D) [52]. The mixture of thiolated HA and thiolated CMC served as the precursor solution, exhibiting a slow gelling time (1.4–7.0 min) because the disulfide bonds crosslinks occurred gradually at physiological temperature. Li et al. constructed thiolated HA and maleilated collagen (Col-MA) with double bonding and a carboxyl group [53]. The mixture of the two solutions could easily be administered through a syringe. After injection, the double bonding in Col-MA and the thiol group in HA underwent the Michael addition under physiological conditions, resulting in the formation of the hydrogel. Wang et al. developed an injectable hydrogel utilizing aminated HA and aldehyde-functionalized β-cyclodextrin through the Schiff base [54]. The aldehyde β-cyclodextrin not only encapsulated the hydrophobic drug but also crosslinked with the aminated HA (Figure 1E). The mixture of β-cyclodextrin and aminated HA could be injected in liquid form and then gelatinized via Schiff base crosslinking. Recently, the DA click reaction between tetrazine and norbornene has been demonstrated to facilitate the delayed formation of crosslinked hydrogels without the need for an external energy input. Koshy and co-workers utilized tetraazine and norbornene to modify gelatin, respectively [55]. The mixture of these two types of modified gelatin allowed for the generation of the spontaneous DA click reaction, leading to the formation of a stable hydrogel in situ within a few minutes after injection. Bi and co-workers synthesized a heat-sensitive furanyl-modified hydroxypropyl chitin [56]. The aqueous mixture of this modified hydroxypropyl chitin and bimaleimide-functionalized PEG can be easily injected into the desired site due to its fluid nature at low temperatures. The hydrogel then forms in situ under physiological conditions, owing to the DA reaction between maleimide and furanyl groups (Figure 1F). Wang et al. introduced furan groups into HA and mixed them with dimaleimide PEG [57]. After injection, a thermally induced DA click reaction occurred between furan groups on HA and the maleimide groups on PEG at 37 °C. The slower kinetics of this reaction persistently improved the hydrogel’s mechanical strength.

Chemical crosslinking methods are easy to control in terms of the flexibility and spatiotemporal precision of the crosslinking process [58]. As a result, injectable hydrogels formed through chemical crosslinking demonstrate enhanced stability and controllability. Generally, injectable hydrogels with chemical crosslinking exhibit higher elasticity compared to physical crosslinked hydrogel, owing to a higher degree of crosslinking [26]. After injection, the robust binding within these hydrogels prevents dilution or diffusion into surrounding fluids, ensuring an effective and lasting filling effect. Nonetheless, the biosafety of some chemically crosslinked hydrogels remains a concern. They often provoke inflammatory responses after injection, which can hinder interactions with host cells and affect patient experiences. Moreover, certain chemical modification procedures are costly and labor-intensive, limiting their broader application.

**Figure 1 pharmaceutics-16-00430-f001:**
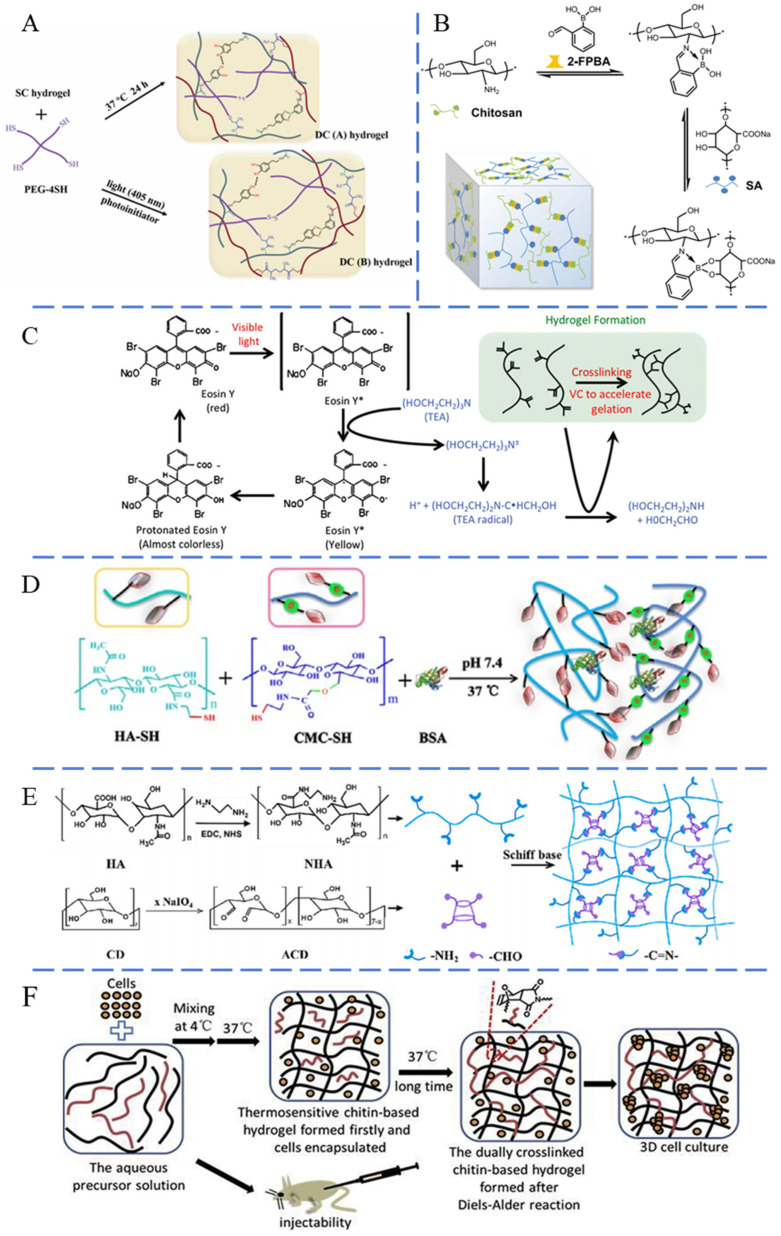
Schematic of chemical covalent crosslinked injectable hydrogel via (**A**) reversable bonding based on Michael addition method [30], (**B**) reversable bonding based on borate imine and borate bonds [45], (**C**) delayed gelatinization based on radical polymerization [48], (**D**) delayed gelatinization based on thiolated HA/CMC [52], (**E**) delayed gelatinization based on Schiff base reaction [54] and (**F**) delayed gelatinization based on Diels–Alder reaction [56]. Reproduced with permission.

### 2.2. Physical Crosslinking

Physical hydrogels have come into study focus due to their ability to self-assemble under specific conditions without the need for crosslinkers [33]. Physical crosslinking, such as hydrogen bonds, hydrophobic interactions, host–guest interactions, ionic crosslinking, and electrostatic interactions, represents weak internal attractive forces. These can temporarily disintegrate under shear forces and reconstitute once the external forces are alleviated. Consequently, physically crosslinked hydrogels possess the reversible capability to transition through a syringe needle in a liquid-like state and subsequently transform into gels at the injection site.

#### 2.2.1. Hydrogen Bonding

Hydrogen bonding is formed between a hydrogen atom and hydrogen bond acceptor with a lone pair of electrons [59]. Hydrogels formed through hydrogen bonding consist of reversible crosslinked networks, where the hydrogen atom interacts with electronegative atoms, such as nitrogen, oxygen, and fluorine. Hydrogen bonding, noted for its dynamic nature, acts as a crosslinking strategy in the development of injectable hydrogels. Additionally, hydrogen bonding can endow hydrogels with self-healing properties, thermoplasticity, and reprocessability. Hydrogen bonding is considerably weaker than covalent and ionic bonds [60,61]. Therefore, it is often combined with other crosslinking mechanisms to enhance the stability of hydrogels. A notable compound, 2-ureido-4[1H]-pyrimidinone (UPy), is widely utilized to facilitate the formation of multiple hydrogen bonds, thereby enabling the establishment of quadruple hydrogen bonding [51]. Zhang and co-workers constructed strong quadruple hydrogen bonding interactions based on UPy moieties to design injectable hydrogel [62]. These hydrogen bonds acted as physical crosslinkers and were broken during the injection. After the injection, the hydrogen bonds reformed to generate the hydrogel. Similarly, Zhao and colleagues fabricated an injectable hydrogel through the polymerization of *N*-acryloyl glycinamide (Figure 2A) [63]. After being extruded from the syringe, the disrupted hydrogel self-healed into integral hydrogel due to the reversible hydrogen bonds.

#### 2.2.2. Hydrophobic Interaction

Hydrophobic interactions represent another commonly used method for creating physically crosslinked hydrogels with injectability. For construction, polymer chains must possess both hydrophobic and hydrophilic segments. The hydrophobic segments tend to aggregate, forming micelles due to hydrophobic effects [64]. These micelles act as the crosslinking points, allowing the polymer chains to form a crosslinked network, thus generating the hydrogel [65]. Hydrophobic interactions can be precisely adjusted by modifying the shape of the hydrophobic areas and the quantity of hydrophobic groups [66,67]. More importantly, these interactions are reversible, endowing the hydrogel with injectable properties. The compact hydrophobic structure is easily assembled, and the hydrophobic association area is rapidly reformed when destroyed. Chiu et al. conjugated hydrophobic palmitoyl groups to the free amine groups on chitosan, creating the polyelectrolyte *N*-palmitoyl chitosan. In an aqueous environment, this compound behaved as a shear-thinning fluid, facilitating its smooth passage through a syringe. [68]. After injection, it rapidly transferred into a hydrogel state in situ at a pH value of 7.4, due to the hydrophobic effect. Liu and co-workers prepared an injectable hydrogel through the physical bonding between hydroxybutyl chitosan (HBC) and silk fibroin (SF) [69]. The hydrophobic blocks on HBC assembled at 37 °C, which caused the aggregation of hydrophobic blocks on SF through hydrophobic interactions, facilitating the transition from a sol to a gel state. The HBC/SF precursor solution gelatinized after the injection for 5–8 min at 37 °C. When the hydrogel is reset at low temperature, it reverses to a flowing liquid within 10 min. Zhao et al. reported an injectable hydrogel through the hydrophobic interactions of four-arm polymers, as illustrated in Figure 2B [70]. The hydrophobic interactions of poly(γ-o-nitrobenzyl-l-glutamate) not only acted as reversible crosslinkers, connecting the star-shaped polymers and facilitating a sol–gel transition during the injection, but also provided a hydrophobic pocket for loading hydrophobic pharmaceuticals.

#### 2.2.3. Host–Guest Interactions

Host–guest interactions involve non-covalent bonding between compounds consisting of host molecules and guest molecules or ions [71]. This non-covalent force is reversable, endowing the corresponding hydrogels with injectable capabilities. Cyclodextrin and its derivates are widely used as the host component for injectable hydrogels, while adamantane, ferrocene, azobenzene, cholic acid, and cholesterol can serve as the guest groups [72,73]. The mechanical strengths of hydrogels based on host–guest interactions can be modulated by controlling the assembly time or by varying the balance between hydrophilicity and hydrophobicity in the hydrogel [74]. Wu et al. synthesized hydrophilic copolymers of PEG and aniline oligomer blocks to serve as the guest molecule for γ-cyclodextrin, constructing an injectable hydrogel (Figure 2C) [75]. The polymer solution underwent a reversible sol–gel transition and could form a hydrogel in situ after injection. Rodell and co-workers synthesized adamantane-modified HA (guest) to couple with cyclodextrin-modified HA (host) [76]. Through the host–guest interaction, the mixture of these two component solutions formed a hydrogel, which exhibited liquid flow under large strain and could reassemble into a network without strain to reform the hydrogel.

#### 2.2.4. Other Physical Interactions

Other physical bonding, such as π-π stacking, ionic electrostatic bonds, and ionic coordination, can also be utilized to create injectable hydrogels. π-π stacking refers to the specific spatial arrangement between aromatic compounds and the weak bonding in the aromatic ring. This phenomenon typically arises between electron-rich and electron-deficient molecules [77,78]. Numerous studies have combined π-π stacking with other interactions to prepare injectable hydrogels. Yang and co-workers found that a solution of adipic acid dihydrazide-modified HA with aldehyde-modified Pluronic F127 (PF127) micelles and dopamine-functionalized oxidized HA could form a hydrogel after injection [79]. The hydrogel network was reinforced by hydrogen bonds and π-π stacking. Ionic interactions are generated through the electrostatic force between molecules of opposite charge. Polymers with opposite charges can be applied to synthesize injectable hydrogels [80]. Sun et al. modified silk fibroin (SF) to produce silk acid (SA) with a carboxylation degree of 9.5%, employing it as a precursor for injectable hydrogels [81]. Under the physiological conditions, SA maintained the solution state within the initial 12 h. With a prolonged time, the SA solution gradually gelatinized due to the strong ionic interaction and hydrophobic interaction, which can be applied for the injectable hydrogel. Rybak et al. prepared a physical hybrid hydrogel using PF127 and calcium ion crosslinked alginate [82]. This hydrogel exhibited thermo-reversible characteristics, remaining in a liquid state at room temperature and transforming into a hydrogel in vivo after injection, due to the thermosensitivity of PF127. Similarly, Nilforoushzadeh and colleagues designed a thermosensitive injectable hydrogel based on poly(n-isopropyl acrylamide) (PNIPAm) and gelatin [83]. Upon injection, the hydrogel absorbed heat from the skin, rapidly undergoing a sol–gel transition in situ. The coordination bond between a metal center and a ligand is established through the donation of electron pairs from a ligand to the metal center. This process is reversible and well suited for the construction of injectable hydrogels [84]. For example, Azadikhah and colleagues developed an injectable supramolecular hydrogel using polyvinyl alcohol (PVA), chitosan, and tannic acid through hydrogen bonding and metal–ligand coordination [85]. Tannic acid molecules could crosslink PVA and chitosan via hydrogen bonds. Furthermore, Fe (III) ions coordinated with the phenolic hydroxyl groups of tannic acid to form an additional network (Figure 2D). This hydrogel exhibited shear-thinning properties and could self-heal to reform hydrogel in situ through the reversible coordination bonds and hydrogen bonds after injection.

**Figure 2 pharmaceutics-16-00430-f002:**
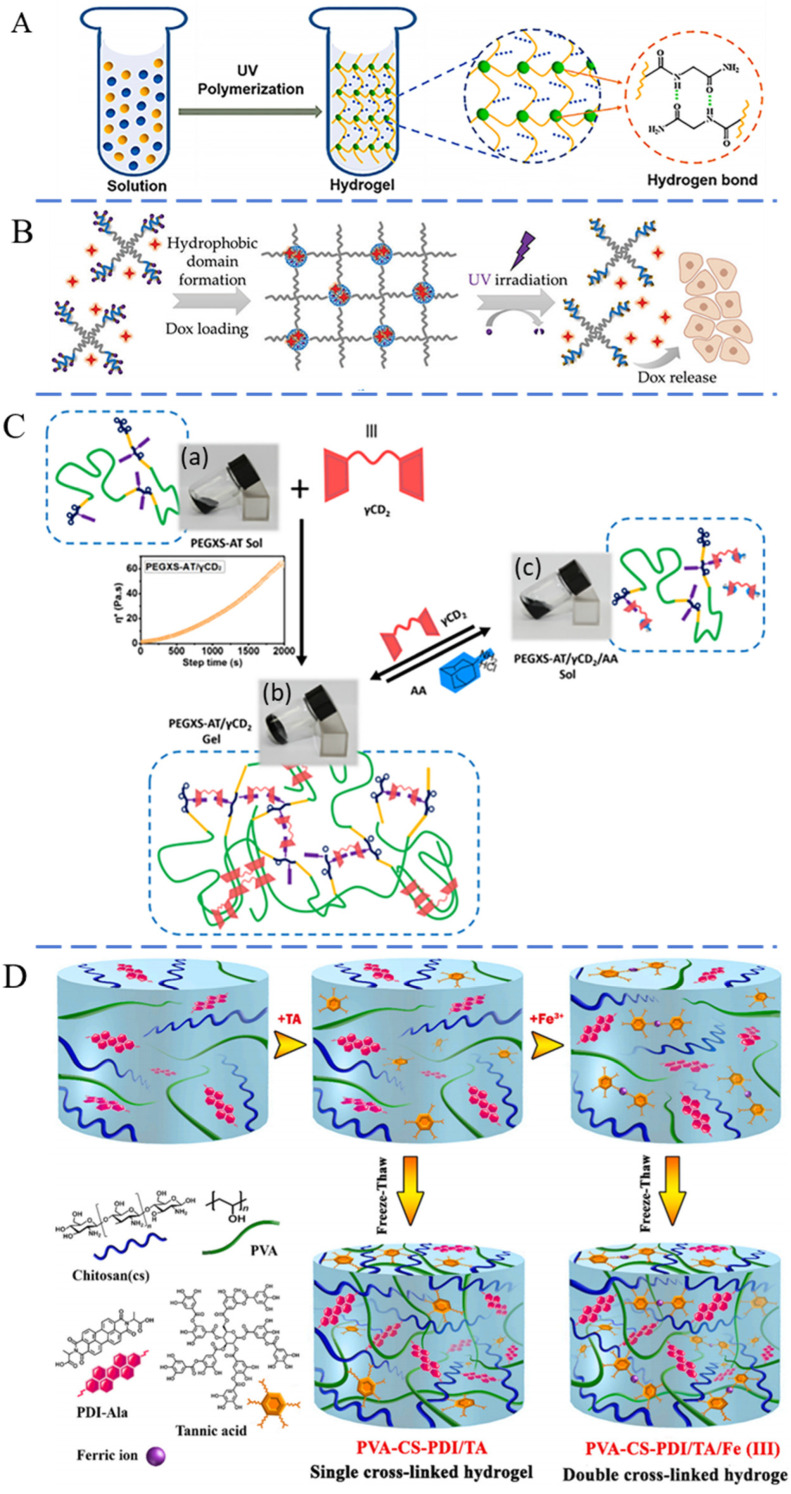
Schematic of physical crosslinked injectable hydrogel via (**A**) reversable bonds based on hydrogen bonding [63], (**B**) reversable bonds based on hydrophobic interactions [70], (**C**) reversable bonds based on host−guest interaction [75] and (**D**) reversable bonds based on multiple physical interactions (including hydrogen bonding and ions coordination) [85]. Reproduced with permission.

The advantages of physical crosslinked hydrogels are apparent. They are easier to prepare and do not require additional crosslinking agents. However, their drawbacks are equally noticeable. These physical interactions are generally weaker than covalent crosslinking, resulting in the poor mechanical properties and low stability of hydrogels, which are significant limitations for filling applications. In addition, the formation of physical hydrogels usually requires certain environmental conditions, such as a suitable pH value or temperature range. Conversely, if certain conditions are exceeded, the hydrogel may not form, or the formed hydrogel structure may be destroyed. Given these considerations, especially for applications in complex subcutaneous environments, there is a growing consensus that hydrogels should not rely solely on a single type of crosslinking. The synergy of multiple reversible dynamic bonds has emerged as a trend in the preparation of various injectable hydrogels.

### 2.3. Biological Crosslinking

Hydrogels prepared through biological methods primarily rely on enzymatic crosslinking. This approach is highly valued in research and has considerable application potential owing to its advantages, such as high efficiency, desirable selectivity, temperate reaction conditions, and excellent biosafety [86,87]. Enzyme-mediated injectable hydrogels can regulate macromolecular composition and modulate enzyme kinetics to control the gelation rate for many biomedical applications. In 1997, Sprinde et al. first crosslinked PEG and lysine-containing polypeptides to generate PEG-based hydrogels using natural glutamine transaminase [88]. This hydrogel exhibited potential for kinetic control over the gelation process, endowing it with promising applications in extrusion and injection. Bi and colleagues developed a novel injectable hydrogel through the enzymatic crosslinking of tyramine-grafted carboxymethyl chitin. This hydrogel, in its precursor liquid form, can be injected and subsequently catalyzed by horseradish peroxidase under physiological conditions to form a tyramine-incorporated hydrogel, as illustrated in Figure 3A [89]. This novel hydrogel exhibited a more desirable mechanical property than commonly encountered with physically crosslinked hydrogels. Tang et al. modified gelatin with tyramine, which could be catalyzed by horseradish peroxidase to produce an injectable hydrogel [90]. Specifically, enzymatic crosslinking reactions can also be used to prepare injectable microgels. Hou et al. crosslinked gelatin with microbial transglutaminase to fabricate microgels (Figure 3B) [91]. After injection, the transglutaminase created bonds between glutamine and lysine residue on gelatin, resulting in the reformation of bulk hydrogels.

Enzymatic crosslinking has addressed certain challenges associated with hydrogel constructed by physical or chemical approaches. Hydrogels with enzymatic crosslinking avoid the use of toxic compounds, and corresponding studies have shown that cells exhibit high proliferation ability within these enzyme-crosslinked hydrogels, demonstrating desirable biocompatibility [92,93]. However, a plethora of substrates presents a significant challenge, necessitating the screening and evaluation of crosslinking enzymes capable of effectively crosslinking diverse substrates.

## 3. Composition of Injectable Hydrogels

The hydrogel can be described as a crosslinked polymer network that confines the flow of internal water. Consequently, the physicochemical properties of the polymer directly affect the hydrogel’s characteristics. Both natural polymers, such as protein and polysaccharide, and synthetic polymers, including polyvinyl alcohol (PVA), polyethylene glycol (PEG), and polylactic acid (PLA), have been utilized to construct injectable hydrogels for tissue filling [94].

### 3.1. Natural Polymer Hydrogels

Natural polymers have been favored in the preparation of hydrogels with injectability for tissue filling owing to their distinct advantages in biosafety and biodegradability. These materials are often regarded as safer options and are more readily approved by the Food and Drug Administration (FDA) for clinical applications [95,96]. Furthermore, natural materials, including HA, SF, collagen, gelatin, and cellulose, possess more potent intrinsic biological functions compared to synthetic polymers. For example, HA, a crucial component of the ECM, offers numerous benefits, such as low immunogenicity, biodegradability, and the promotion of cell proliferation. Consequently, in the field of aesthetics, HA-based hydrogels are extensively utilized for enhancing cutaneous contours and correcting depressions [97,98]. Collagen, another ECM component, comprises a three-helices structure and is able to assemble under physiological conditions in vitro to form collagen fibrils with the same hierarchical structure in natural tissues [99]. It has been widely utilized in tissue filling due to its ability in promoting cell proliferation [100]. SF, a protein produced by silkworms, is abundantly available and relatively inexpensive. SF exhibits optimal biosafety and remarkable biological functions, including the promotion of cell proliferation and angiogenesis, making it an ideal candidate for injectable fillers [101,102]. Gelatin, the hydrolysate of collagen, partly retains the primary structure of natural collagen. It possesses numerous advantages, such as biodegradability, renewability, cost-effectiveness, and biocompatibility. Crosslinked pure gelatin or gelatin combined with other natural/synthetic polymers has been fabricated into numerous injectable hydrogels for soft tissue filling [103]. Although these natural polymers have some shortcomings, such as rapid degradation and low mechanical strength, adjusting the concentration or introducing crosslinking on the main chain may overcome these disadvantages.

### 3.2. Synthetic Polymeric Hydrogels

Certain synthetic polymers, recognized for their favorable biosafety profiles, also hold potential in the creation of injectable hydrogel for tissue filling. For example, PVA, a hydrophilic polymer, is relatively inexpensive and has been utilized in both soft and bone tissue engineering [104]. Compared to natural polymers, synthetic polymers offer distinct advantages, such as improved mechanical properties, slower degradation rates, and the capability for customization to achieve specific physicochemical properties. However, they also have notable disadvantages, such as biological inertness [105,106]. PLA and polylactic-co-glycolic acid (PLGA) are considered optimal materials for hard tissue repair due to their promising biosafety profile, slow degradation, and high mechanical strength [107]. PEG undergoes degradation, primarily through oxidation rather than hydrolysis, resulting in a slow degradation rate at the injection site and a prolonged filling effect [108]. Additionally, synthetic polymers frequently cooperate with natural polymers to compensate for each other’s shortcomings.

## 4. Properties of Injectable Hydrogels

The injectable hydrogel must exhibit high fluidity to ensure its passage through a syringe. Furthermore, once injected, the hydrogel should remain stable and possess properties that meet the requirements of its biomedical application. Therefore, in the development of an injection system for filling, hydrogels need to be customizable and equipped with characteristics relevant to their application, such as appropriate mechanical properties, adjustable degradability, and biological functionality.

### 4.1. Mechanical Strength of Injectable Hydrogels

After injection, possessing the appropriate mechanical strength is crucial for the hydrogel’s success as a filler. The mechanical strength of different soft tissues is diverse. For instance, the elastic modulus of muscle and skin is in the ranges of 10–18 kPa and 0.2–2 kPa, respectively [109,110,111]. Thus, the mechanical strength of injectable hydrogels should be tailored to match the targeted tissue. Gold and colleagues developed crosslinked methylcellulose (MC) hydrogels specifically for soft tissue [112]. The equilibrium modulus of the composite hydrogel ranged from 1.4 to 5.3 kPa, comparable to human adipose tissue. Furthermore, the gelation time of the MC hydrogels met the International Organization for Standardization (ISO) standard for injection materials. Lee et al. crosslinked HA with auto-oxidized gallic alcohol to prepare a hydrogel [113]. The shear modulus of the hydrogel was about 2.0 kPa, which was 7.6-fold (0.27 kPa) of commercial Restylane, a HA filler product (Q-Med, Sweden). More importantly, the elastic modulus could be maintained at 0.6–0.7 kPa in vivo for 1 year after injection.

### 4.2. Degradability of Injectable Hydrogels

Synthetic permanent fillers have been associated with complications, such as chronic inflammation, and can only be safely removed through surgical excision [114]. In contrast, biodegradable fillers derived from natural polymers do not require removal due to their ability to decompose naturally. The degradation rate of these fillers is intricately linked to their mechanical strength, playing a critical role in the tissue filling process. A hydrogel that decomposes too rapidly may prematurely lose its mechanical strength, potentially leading to an unsatisfactory filling effect due to the challenged cellular infiltration. Gold and colleagues developed MC hydrogels through a polymerization process using ammonium persulfate and *N*, *N*, *N*′, *N*′-tetramethylethylenediamine [115]. To evaluate its sensitivity to cellulase activity, the MC hydrogel was immersed in a cellulase solution. With increased exposure to the corresponding enzyme, the edges of the hydrogel progressively merged, and the structure became more amorphous, ultimately degrading completely within 48 h. This finding is comparable to studies where hyaluronidase was exhibited to decompose HA filler within 72 h [116,117]. Similarly, Hong et al. designed an injectable dopamine-modified HA hydrogel filler with self-crosslinking. This hydrogel was immerged in hyaluronidase solution for 4 weeks, and its weight was reduced to 15–19% of the initial weight [118]. For comparison, these HA hydrogels without enzyme treatment remained at their original weight for 4 weeks. Furthermore, after 4 weeks of subcutaneous injection into mice, the hydrogel remained well integrated, with the residual volume of the filler approximating 80% of its initial volume.

Degradation typically occurs more slowly in harder hydrogels because these hydrogels have a denser network, which prolongs the degradation time. Liu et al. prepared a multi-functional HA and human-like collagen (HLC) hydrogel with injectability for tissue filling [119]. The incorporation of HLC significantly enhanced the hydrogel’s elastic modulus from 0.05 to 0.21 MPa. After being soaked in hyaluronidase for 6 weeks, the hydrogel containing HLC had a residual weight of 62.4%, markedly higher than that of the hydrogel without HLC (18.4%). Furthermore, after undergoing simultaneous degradation by collagenase I and hyaluronidase, 45.4% of the hydrogel remained. In vivo injection experiments indicated that the residual weight ratios of the hydrogel, both without and with HLC, were 14.1% and 54.6% in Kunming mice, and 8.6% and 42.4% in New Zealand rabbits, respectively.

### 4.3. Biological Function of Injectable Hydrogels

The introduction of fillers can potentially induce adverse effects, such as rejection. Complications associated with absorbable fillers typically resolve spontaneously within a few months, but non-degradable fillers may lead to long-term physical and psychological harm [120]. Therefore, the ideal injectable filler should not only be stable at the implantation site to ensure satisfactory aesthetic outcomes without migration but also exhibit exceptional biosafety to minimize the immune response in vivo. Numerous studies have detailed in vitro cytotoxicity assays and utilized various indicators, such as cell proliferation, collagen deposition, and angiogenesis, to evaluate the in vivo immune response post-injection for biosafety [121,122,123]. Degradable hydrogels provide space for cell stretching and proliferation, and studies have shown that between different hydrogel matrices, an increasing 3D cell speed with increasing pore size was apparent. Cell behavior, such as diffusion, migration, and proliferation, could be modulated by the physical properties (e.g., stiff and soft substrates) and structure of hydrogels [124,125]. For example, the physical properties of the cellular environment can directly affect epithelial growth and guide cell migration [126]. Kim et al. incorporated the basic fibroblast growth factor (bFGF) into photo-crosslinked HA hydrogel via the thiolene reaction and evaluated its biofunctions for fibroblast proliferation and migration [127]. HA bound with the CD44 receptor, which activated various signaling pathways related to cell proliferation, adhesion, and migration. In vitro experiments exhibited that the hydrogel with bFGF significantly enhanced cell proliferation and migration. The combination of bFGF and HA hydrogels could mediate cell migration through the sustained release of bFGF from the HA hydrogel matrix and the interaction of CD44 with HA. Similarly, Brown et al. prepared injectable SF microparticle-based fillers and then crosslinked them with HA (SF-HA) [128]. Compared with CaHA-CMC, a commercial particle filler, SF-HA exhibited no negative responses on macrophages and recruited fibroblasts to the site of remodeling. The regenerated fibrous tissue infiltrated around the implanted particles, resulting in the deposition of interstitial fibrous tissue, thereby achieving the desired filling effect. Hahn and co-workers prepared a new filler based on HA hydrogel particles [121]. After subcutaneous injection, there was no significant inflammatory response or abnormality in skin thickness, indicating excellent in vivo biosafety. At 4 to 8 weeks after the injection, fibroblasts surrounded the filler material and then formed a lattice structure by fibrogenesis and angiogenesis. The process of angiogenesis around the filler material provided a framework for autologous tissue to fill the space with the gradual degradation of the filler materials. Moreover, at 12 weeks post-injection, both collagen content and elastin fibers exhibited a significant increase compared with the saline-treated control group, suggesting the filler’s biofunctions in stimulating extracellular matrix production and angiogenesis. Bi et al. developed an injectable hydrogel by producing tyramine-modified CMC and subsequently using enzymatic catalysis [89]. The results from the CCK-8 assay and the live/dead staining confirmed that the hydrogel had excellent biosafety. Although there was an initial increase in the number of inflammatory cells within the first 4 days post-injection, this inflammation gradually diminished and completely resolved over the following 2 weeks. Throughout the experimental period, there were no obvious signs of edema, hyperemia, or tissue necrosis.

## 5. Injectable Hydrogels for Soft Tissue Fillers

Injectable hydrogels have gained widespread popularity in cosmetic minimally invasive procedures for filling and regenerating tissue. A diverse array of materials, including HA, SF, collagen, CMC, alginate, chitosan, PEG, PLGA, PLA, poly(methyl methacrylate), and polycaprolactone, were adopted for the synthesis of hydrogel with injectability.

### 5.1. HA-Based Injectable Hydrogels

HA, a natural glycosaminoglycan, is the main ingredient of the ECM. HA is known for its excellent viscoelasticity, good biosafety, and biodegradability [129]. Based on these properties, HA hydrogel with injectability was extensively used in both cosmetic and medical fields, becoming the most commercially popular filler with continuously growing demand. Despite HA being the preferred choice for soft tissue filling, its poor mechanical strength and undesirable stability limit its usage [118,130]. The strength and stability of HA can be enhanced through crosslinking or by adding other biomaterials to form hydrogels [131]. Lee et al. synthesized an HA hydrogel with self-crosslinking ability and applied it as the injectable dermal filler [113]. A corresponding experiment confirmed that this filler outperformed commercial products (such as Restylane) in terms of injectability, tissue adhesion, and lasting retention. Additionally, this filler is capable of encapsulating drugs or growth factors, thus promoting cell proliferation. Kim and co-workers reported the development of an injectable HA hydrogel loaded with bFGF for use as a dermal filler [127]. After injection, this hydrogel was able to sustain the filling effect for an extended period, with bFGF inducing cell proliferation and migration. Similarly, Hong and colleagues developed a dopamine-modified HA hydrogel with desirable injectability and biocompatibility [118]. This HA hydrogel demonstrated a compression recovery of up to 95% and needed an injection force of about 5 N. For comparison, commercial filler products like Restylane required an injection force of about 20 N, indicating significantly inferior injectability.

Although certain hydrogels exhibit excellent mechanical properties and elasticity, their inherent characteristics may not render them directly suitable for injection. To address this issue, these hydrogels can be processed into microgels, thereby creating fillers with enhanced injectability. Hong et al. developed bulk HA hydrogels, which were then cut into micrometer-scale particles by a mesh [132]. These micro-particles could be injected after dispersion in solution. Regulating the crosslinking degree could adjust the strength of hydrogel to meet the requirements of different injection sites (lips and eyes). Similarly, Hyunsuk and co-workers fabricated HA-poly(nucleotide)/PLA particles for dermal filler using a microfluidic system [98]. This novel filler was nontoxic and could maintain the stable volume after the injection for 24 weeks.

### 5.2. SF-Based Injectable Hydrogel

SF, extracted from silkworm cocoons, has excellent biosafety, controllable biodegradability, and low immunogenicity [133,134]. SF hydrogels have found wide application in biomedical fields, including tissue scaffold, drug delivery, and wound dressing [135]. Particularly, SF hydrogels with injectability could be applied for soft tissue filling. Zeplin et al. crosslinked SF to prepare a hydrogel, which could pass through a 27G needle [136]. After injection in mice and minipigs, no postoperative complications were observed, and the SF hydrogel fully degraded within 90 days. Hybrid hydrogels based on SF and other natural polymers also demonstrate potential for injectable applications. Liu and colleagues prepared a thermosensitive HBC/SF hybrid hydrogel through physical bonding [69]. The HBC/SF mixture transitioned into a hydrogel state within several minutes and could rapidly gelatinize in situ after injection. The biosafety of the HBC/SF hydrogel was obviously better than that of pure HBC, making it a promising candidate for use as an injectable tissue filler.

SF can also be fabricated into microparticles for injection. Brown et al. chemically crosslinked SF and HA to prepare injectable SF/HA microparticles [128]. These microparticles exhibited mechanical properties that mimic the modulus of natural tissue. The injection force required for the SF/HA microparticles was lower than that for the commercial filler (Prolaryn Plus, Merz Pharma, Frankfurt, Germany) (Figure 4A). After injection, the SF/HA microparticles exhibited desirable biosafety and a slow decomposition rate, and they could facilitate tissue regeneration, as depicted in Figure 4B. Similarly, our research group developed bulk SF hydrogels and processed them into microparticles using meshes [101]. The diameter of microparticles could be modulated by the mesh size to fit different needles. After injection, these particles did not induce an obvious inflammatory response and could promote angiogenesis and collagen deposition. Furthermore, 12 weeks later, the filling effect was still obvious, suggesting a lasting filling effect.

### 5.3. Collagen-Based Injectable Hydrogels

Collagen, a primary component of the ECM, plays a crucial structural role in maintaining tissue architecture [137]. In aesthetic medicine, in 1981, with the development and FDA approval of the first collagen filler, Zyderm (Inamed Corporation, Santa Barbara, CA, USA), the preparation and optimization of collagen hydrogels for tissue filling became a hot research topic [138]. Dermicol-P35 (Evolence, Ortho Dermatologics, Skillman, NJ, USA), Artecoll (Canderm Pharma, Saint Lorent, QC, Canada), and Cymetra (Lifecell Corp, Branchberg, NJ, USA) have been shown to achieve a high degree of correction in depressed acne scars, showing their effectiveness without adverse events [139,140,141]. Liu et al. developed a novel injectable hydrogel based on HLC and CMC [142]. This hydrogel possessed desirable water absorption and biosafety. After injection, the hydrogel degraded slowly and could still be observed 28 days post-implantation. Similarly, Li and colleagues prepared multi-functional injectable hydrogels based on pullulan and HLC [100]. The incorporation of HLC into pullulan hydrogels significantly enhanced the elastic modulus and cell adhesion properties compared to pure pullulan hydrogels. After injection, the pullulan/HLC hydrogel did not provoke notable inflammation, and its degradation period extended beyond six months, indicating a lasting filling effect. Liu et al. designed HA and HLC hydrogels with injectability for use as soft tissue fillers [119]. After injection, the hydrogel with a higher HLC ratio exhibited slight inflammation, and its degradation time extended beyond 4 months. Ding et al. prepared poly (D, L-lactide) microspheres/collagen hydrogel, and fibroblasts were encapsulated in this hydrogel [122]. This composite was heat-sensitive, enabling a sol–gel transition that endowed the hydrogel with injectable properties. After injection, it was not only stable, providing a lasting filling effect, but also significantly stimulated collagen regeneration and promoted the formation of new connective tissue.

### 5.4. CMC-Based Injectable Hydrogels

CMC, a plant-derived polysaccharide, received FDA approval for biomedical applications due to its desirable biocompatibility and reduced immune response [143,144]. Furthermore, the lack of cellulase in humans ensures the stability of CMC after injection [145]. Some commercial fillers, such as Laresse and Sculptra, combine CMC with other materials (polyethylene oxide, hydroxyapatite (HAP), and PLA for dermal injectable fillers) [146,147]. Varma et al. developed a redox polymerized CMC hydrogel with a tunable equilibrium modulus ranging from 2.0 to 9.2 kPa, similar to that of natural soft tissues (human fat, mammary gland, and nucleus pulposus tissue) [148]. The rheological properties of this CMC hydrogel met the ISO standard for injectable materials. Nonetheless, the absence of in vivo experimental data necessitated further research to assess the hydrogel’s filling efficacy and biosafety. Choi et al. prepared injectable hydrogels based on levan, PF127, and CMC [149]. Owing to the thermosensitivity of PF127, this hydrogel could remain in a liquid state at lower temperature for injection but reform the hydrogel state at 37 °C. After injection, the stable filling duration of this hydrogel was longer than that of PF127 or HA hydrogel alone. More importantly, this hydrogel could promote collagen deposition (Figure 5A,B) and possessed desirable anti-wrinkle efficacy (Figure 5C,D).

### 5.5. Other Injectable Hydrogels

There are other hydrogels with injectability for soft tissue filling, in addition to the traditionally used injectable hydrogel fillers based on HA, SF, collagen, and CMC. Xiao et al. copolymerized poly(amidoamine) and *N*-isopropylacrylamide to prepare injectable hydrogels with temperature sensitivity [150]. After a 6-month post-injection period in rats with skin defects, there was a notable increase in the thickness of both skin and muscle in the affected area. Crucially, the hydrogel completely degraded, and the defect was filled with adipocytes and immature adipocytes, indicating great potential for treating skin defects. Pan and colleagues engineered a temperature-sensitive hydrogel with injectability from poly (D, L-lactide)/PEG [108]. This filler could be injectable at room temperature and become a hydrogel at 37 °C. Although inflammatory infiltration was observed around the injection site after 5 weeks, by 9 weeks, the inflammation disappeared, and regenerated collagen fibrils filled the injection site.

In addition to degradable fillers, researchers are also interested in novel semi-permanent injectable hydrogel fillers due to their potential for improved longevity and collagen deposition [151,152]. For example, HAP, the primary ingredient of bone, is a commonly used biocompatible bioceramic that can stimulate collagen deposition after injection into the body [152,153]. The degradation rate of HAP is slow, allowing it to be hybridized with polymers to construct semi-permanent injectable hydrogels. Hwang et al. incorporated 1% HAP into a levan hydrogel to prepare a semi-permanent dermal injection filler [154]. This hybrid hydrogel showed an obvious sol–gel transition as the temperature increased from 4 to 37 °C, providing the hydrogel with desirable injectability. The incorporation of HAP facilitated the proliferation of dermal fibroblasts. After injection for 8 weeks, the hydrogel retained a residual volume of 20~30%, indicating a durable filling effect.

## 6. Conclusions and Outlook

Tissue filling is a multifaceted and dynamic mechanism, aimed at restoring absent cellular structures and layers of tissue. Over the past few decades, substantial efforts have been dedicated to innovating new methodologies for soft tissue augmentation. Consequently, a diverse array of injectable systems has been reported in the literature for tissue filling. Notably, natural biomaterials have been prominently featured in injectable tissue fillers, owing to their bioactive properties inherent in hydrogels derived from these materials. Despite this advantage, a significant drawback is their limited control over mechanical characteristics. In contrast, synthetic polymer-based hydrogels are celebrated for the customizable strength and adjustable decomposition rates. However, they often fall short in terms of biological functionality. To bridge this gap, various crosslinking techniques, including enzymatic catalysis, have been explored. The combination of multiple crosslinking mechanisms is becoming a growing trend to achieve enhanced physical properties and biological functionality. These methods facilitate the integration of natural and synthetic polymers into composite systems, thereby enhancing both the biological efficacy and mechanical robustness of injectable hydrogels. This approach provides a promising alternative for designing ideal injectable hydrogels for tissue filling.

Although numerous injectable hydrogels have been developed for tissue filling, our ability to synthetically replicate the complexities of native soft tissues is still unrefined at best. Currently, the main challenge of injectable hydrogel as a filler is the shortage in systematic research about the relationships among the polymer nature, crosslinking mechanism, the three-dimension microenvironment of hydrogel, cell responses, and the filling effect. This issue results in the problem that existing commercial fillers cannot permanently enhance skin contours and correct depressions, leading to a not entirely satisfactory experience for patients. An ideal injectable medical hydrogel should satisfy several criteria: (1) the conditions and duration of gelation should ensure the stability of hydrogel aligning with the mechanical properties of the surrounding tissue, (2) the rate of biodegradation should align with the pace of tissue healing and regeneration, and (3) the microstructure should support tissue regeneration.

Despite the numerous advantages of injectable hydrogels for tissue filling, certain limitations still require further investigation. Firstly, integrating different polymers with various functionalities, such as enhanced mechanical strength, controlled degradation rates, and improved biological performance, remains a complex challenge. Secondly, for clinical applications, the development and optimization of hydrogel should effectively address clinical challenges, and issues related to preservation techniques must be resolved. Lastly, understanding the mechanisms by which hydrogel induces tissue augmentation is crucial for improving tissue regeneration and achieving promising filling effects, and, thus, these mechanisms should be thoroughly revealed. This review aims to serve as a valuable guide, offering a comprehensive overview to materials scientists, clinicians, and the wider research community interested in the development and application of polymeric injectable hydrogels for tissue filling.

## Figures and Tables

**Figure 3 pharmaceutics-16-00430-f003:**
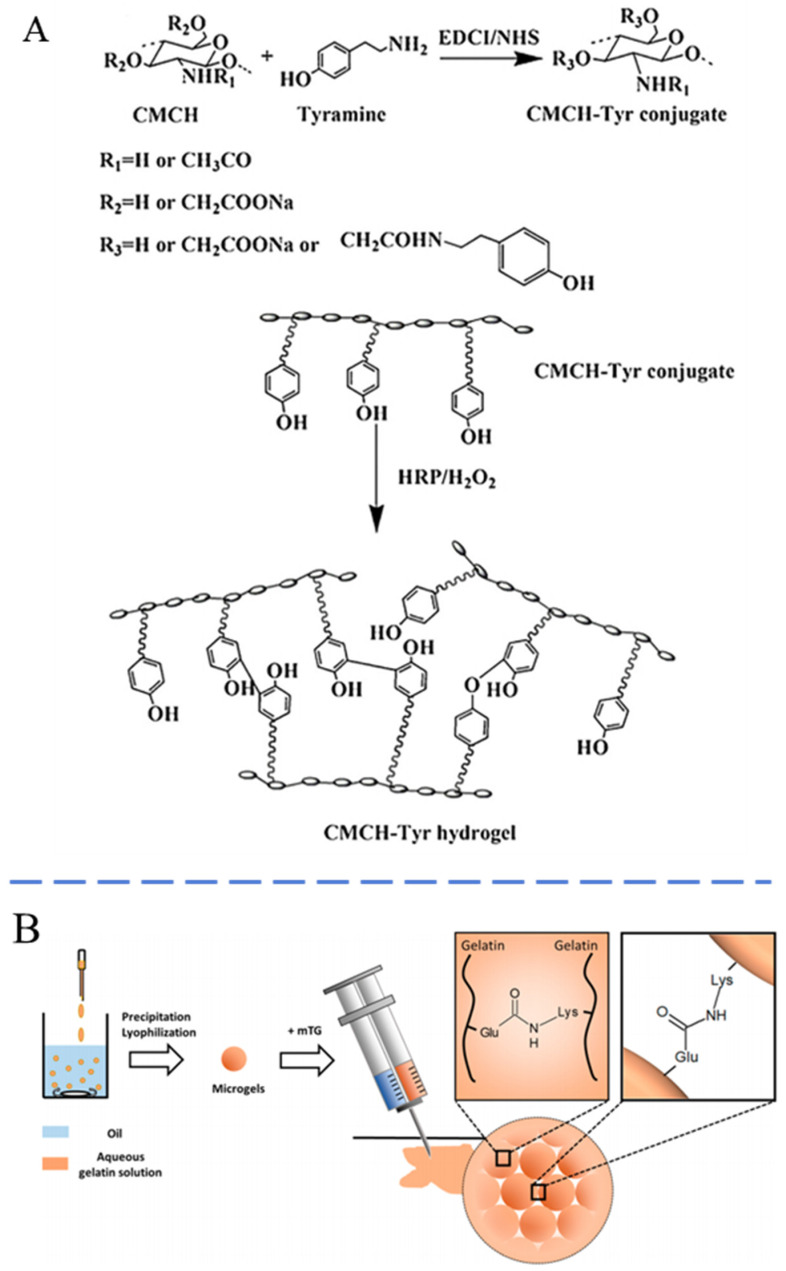
Schematic of (**A**) biological crosslinked injectable hydrogel via delayed gelatinization based on the catalyzation by horseradish peroxidase [89] and (**B**) injectable microgel crosslinked by transglutaminase [91]. Reproduced with permission.

**Figure 4 pharmaceutics-16-00430-f004:**
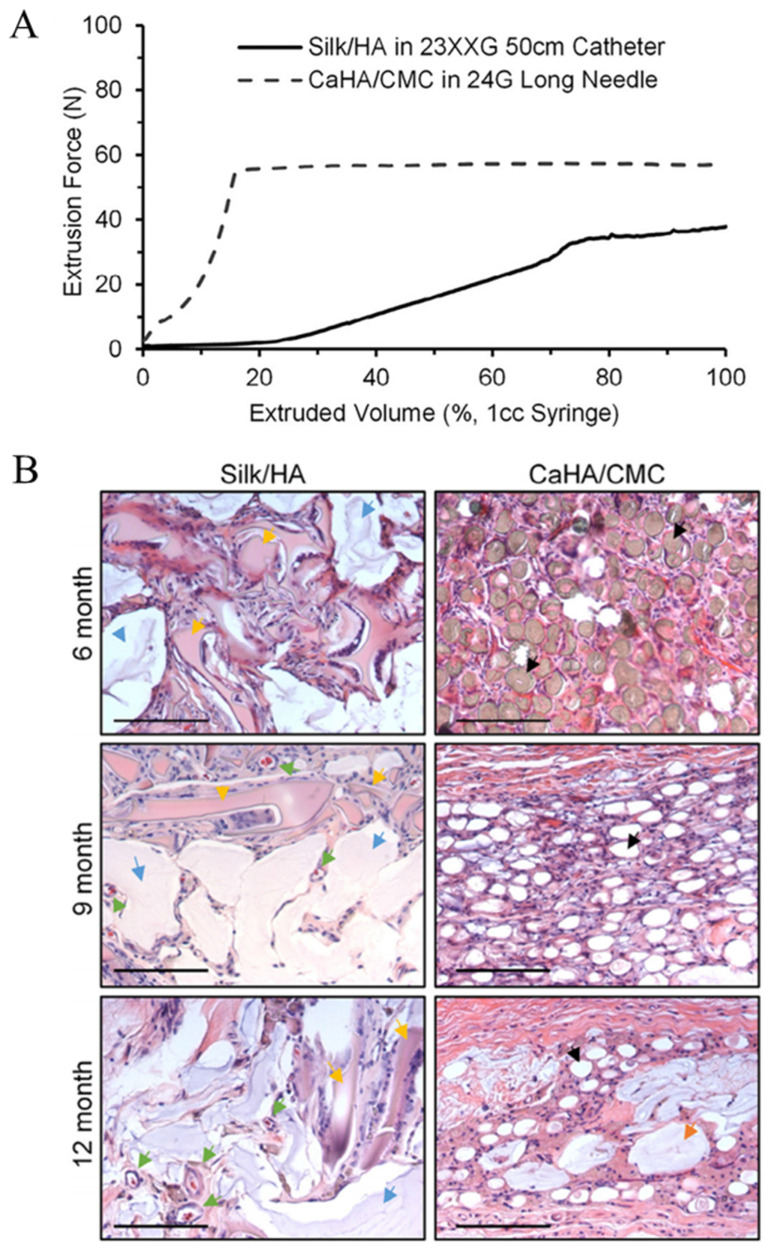
(**A**) The injection force of SF-HA and CaHA-CMC measured with a crosshead speed of 13 mm/min. (**B**) Hematoxylin and eosin staining (H&E) of tissue round the SF-HA and CaHA-CMC injected site (scale bar = 125 µm). Cellular infiltration, predominantly comprising macrophages and giant cells responsible for the enzymatic degradation of silk protein, is observed in proximity to silk particles (yellow arrows). Cross-sections of silk-HA exhibit vascularity (green arrows), within the tissue ingrowth. Areas of HA (blue arrows) demonstrate cell occlusion and undergo collapse during histological processing. Similarly, CaHA-CMC facilitates the infiltration of macrophages and giant cells in areas adjacent to CaHA particles (black arrows). Sections from both 9- and 12-month intervals were subjected to decalcification prior to staining, leading to the formation of "ghost" regions where CaHA particles were once located. Analogous to HA, CMC (orange arrows) exhibits cell occlusive properties. Reproduced with permission [128].

**Figure 5 pharmaceutics-16-00430-f005:**
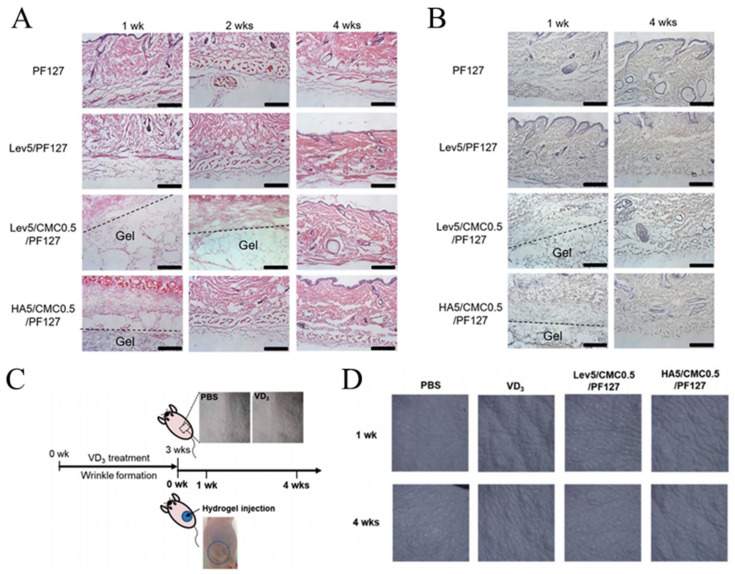
(**A**) H&E staining images and (**B**) immunostaining images of skin tissues injected with the PF127, levan/PF127, HA/CMC/PF127 and levan CMC/PF-127 (scale bar = 200 µm). (**C**) Schematic image for the construction of the hairless mice model with the skin wrinkling and the schedule for injection. (**D**) Skin surface images after injection with different hydrogel. Reproduced with permission [149].

## Data Availability

The data presented in this study are available on request from the corresponding author.

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
