# Peer review of "Synthesis and Properties of Injectable Hydrogel for Tissue Filling"

_pharmaceutics, 2024, doi:10.3390/pharmaceutics16030430_

Round 1

Reviewer 1 Report

Comments and Suggestions for Authors

The present document focuses on a current topic that concerns the use of injectable hydrogels for tissue filling. It summarizes the most recent research work mainly focusing on the physicochemical properties, biological functions and mechanism of action of these hydrogels. The development of new methods for synthesizing these composites based on natural and/or synthetic polymers could constitute a promising alternative for designing a more efficient hydrogel in the tissue filling process. The article is structured with an understandable scientific approach. Sources and reference citations are consistent with the topic discussed. The work arouses interest ; it could serve as a reference for researchers and consequently contribute to the development of materials science.

Nevertheless, here are some comments:

- Some figures are less readable!

- The mechanisms of action of this type of hydrogels are not detailed enough!

- Lack of additional details on possible prospects !

Comments on the Quality of English Language

The level of the English language is generally of good quality. However, it is advisable to avoid language phrases that may affect the meaning of the idea.

Author Response

The present document focuses on a current topic that concerns the use of injectable hydrogels for tissue filling. It summarizes the most recent research work mainly focusing on the physicochemical properties, biological functions and mechanism of action of these hydrogels. The development of new methods for synthesizing these composites based on natural and/or synthetic polymers could constitute a promising alternative for designing a more efficient hydrogel in the tissue filling process. The article is structured with an understandable scientific approach. Sources and reference citations are consistent with the topic discussed. The work arouses interest; it could serve as a reference for researchers and consequently contribute to the development of materials science.

Nevertheless, here are some comments:

Thank you very much for your high evaluation and constructive comments on our manuscript. The manuscript was carefully revised according to your comments. The point-by-point responses were listed below.

  1. Some figures are less readable!

Thank you for the critical question. Following the reviewer’s suggestion, we have revised the figures in the revised manuscript so that readers can see the content clearly.

  1. The mechanisms of action of this type of hydrogels are not detailed enough!

Thank you for the critical question. Following the reviewer’s suggestion, we have made detailed changes to the mechanisms of action of injectable hydrogels in the revised manuscript.

  1. Lack of additional details on possible prospects!

Thank you for the critical question. Following the reviewer’s suggestion, we have made detailed revisions in the outlook section to enhance the discussion of injectable hydrogels for tissue filling.

The level of the English language is generally of good quality. However, it is advisable to avoid language phrases that may affect the meaning of the idea.

Thank you for the critical question. We have edited the language in the revised manuscript.

Reviewer 2 Report

Comments and Suggestions for Authors

In this manuscript, Xie et. al reviewed the use of injectable hydrogels for tissue engineering applications.  I believe improvement of the work is necessary to be published in Pharmaceutics journal. Below I am listing my points for this improvement.

 1) Figure 1,2 very small and very hard to be seen, they should be redrawn to be seen more clearly.

2) Using words such as N-carboxyethyl (116) or N, N, N’, N’-tetramethylethylenediamine (402) N’s should in italic form

3) Throughout the manuscript physical noncovalent bonds/crosslinking term is used, physical already means that interaction is noncovalent, therefore I would choose one of them.

4) with a pH value if 7.4 (243) here if should be of I guess ?

5) Authors should not make incorrect generalizations for examle in section 2.2.2 in sentence 238, authors say that these hydrogels are always reversible, however it may not be. For example, in the case of silk hydrogels they are not reversible.

6) Authors may consider adding some extra references to the natural polymers section such as;

Preparation, properties, and applications of gelatin-based hydrogels (GHs) in the environmental, technological, and biomedical sectors, International Journal of Biological Macromolecules Volume 218, 1 October 2022, Pages 601-633

Self-assembled silk fibroin hydrogels: from preparation to biomedical applications, Mater. Adv., 2022, 3, 6920-6949

Comments on the Quality of English Language

it is ok

Author Response

Reviewer 2

In this manuscript, Xie et. al reviewed the use of injectable hydrogels for tissue engineering applications.  I believe improvement of the work is necessary to be published in Pharmaceutics journal. Below I am listing my points for this improvement.

Thank you very much for your high evaluation and constructive comments on our manuscript. The manuscript was carefully revised according to your comments. The point-by-point responses were listed below.

  • Figure 1,2 very small and very hard to be seen, they should be redrawn to be seen more clearly.

Thank you for the critical question. Following the reviewer’s suggestion, we have revised the figures in detail in the revised manuscript so that readers can see them more clearly.

2) Using words such as N-carboxyethyl (116) or N, N, N’, N’-tetramethylethylenediamine (402) N’s should in italic form.

Thank you for the critical question. Following the reviewer’s suggestion, we have changed the “N” to italics in the revised manuscript.

3) Throughout the manuscript physical noncovalent bonds/crosslinking term is used, physical already means that interaction is noncovalent, therefore I would choose one of them.

Following the reviewer’s suggestion, we have modified “physical noncovalent bonds/crosslinking” into “physical bonds/crosslinking” in the revised manuscript.

4) with a pH value if 7.4 (243) here if should be of I guess ?

Thank you for the critical question. We have corrected “with a pH value if 7.4” into “with a pH value of 7.4” in the revised manuscript. (Line 276)

5) Authors should not make incorrect generalizations for examle in section 2.2.2 in sentence 238, authors say that these hydrogels are always reversible, however it may not be. For example, in the case of silk hydrogels they are not reversible.

Thank you for the critical question. In section 2.2.2, physical crosslinked hydrogels prepared by hydrophobic interactions are generally reversible. Liu and co-workers prepared the silk hydrogel with thermo-sensitivity, which could be transformed into a gel in a short time at 37 °C and then changed into a sol after resetting the low temperature to achieve its reversibility. We have modified the description to explain the mechanism for reversibility in detail in the revised manuscript.

6) Authors may consider adding some extra references to the natural polymers section such as;

Preparation, properties, and applications of gelatin-based hydrogels (GHs) in the environmental, technological, and biomedical sectors, International Journal of Biological Macromolecules Volume 218, 1 October 2022, Pages 601-633

Self-assembled silk fibroin hydrogels: from preparation to biomedical applications, Mater. Adv., 2022, 3, 6920-6949

Thank you for the critical question. Following the reviewer’s suggestion, we have added the suggested references to the natural polymers section in the revised manuscript.

Reviewer 3 Report

Comments and Suggestions for Authors

The manuscript provides a comprehensive overview of injectable hydrogels for tissue filling, particularly in the context of addressing tissue aging and cosmetic applications. It effectively set the stage by highlighting the importance of tissue regeneration, the challenges associated with traditional surgical methods, and the promising role of injectable hydrogels. However, there are a few aspects that could be further addressed or clarified:

(1) The introduction is informative but quite dense. Consider breaking down complex sentences into simpler ones to enhance readability. Provide clear transitions between different sections, ensuring a smooth flow from the introduction to the specific focus on injectable hydrogels.

(2) While the introduction provides a broad context, it would be beneficial to introduce the specific types of injectable hydrogels earlier and elaborate on their unique properties. 

(3) It would be valuable to include a dedicated section on smart hydrogels in the manuscript, discussing their unique characteristics and applications. Additionally, citing relevant works, such as the studies (https://doi.org/10.3390/ijms23073665) and (https://doi.org/10.1021/acsbiomaterials.0c00988), would strengthen the discussion on recent advancements in smart hydrogel research and their potential contributions to the field of injectable hydrogels for tissue filling.

(3) The manuscript mentions "9 unique properties" but doesn't enumerate or explain them in the abstract. Consider providing a concise list or brief elaboration for clarity.

(4) The manuscript mentions intrinsic and extrinsic causes of tissue aging. It would be valuable to include a brief discussion on how injectable hydrogels specifically target or address these causes at a cellular or molecular level.

(5) Emphasize the advantages and disadvantages of different types of injectable hydrogels, as mentioned in the latter part of the introduction, to give readers a clearer understanding of the trade-offs involved in their use.

(6) Elaborate more on the mechanisms of action and classification of injectable hydrogels. For example, discuss how different types of cross-linking affect the stability, mechanical properties, and biocompatibility of hydrogels.

(7) Expand on the biological functions of injectable hydrogels and their therapeutic potential. How do these hydrogels interact with host cells, and what role do they play in tissue regeneration beyond mechanical support?

(8) The abstract mentions recent advancements but does not delve into specific recent studies or breakthroughs. Include more recent examples to highlight the current state of research.

(10) Explicitly mention some of the prevailing challenges in the field to give readers a realistic view of the limitations and areas that need further exploration.

(11) Consider adding figures or illustrations to aid in understanding the concepts discussed, especially in the section about dynamic covalent chemical bonding.

Comments on the Quality of English Language

Minor editing of the English language is required!

Author Response

The manuscript provides a comprehensive overview of injectable hydrogels for tissue filling, particularly in the context of addressing tissue aging and cosmetic applications. It effectively set the stage by highlighting the importance of tissue regeneration, the challenges associated with traditional surgical methods, and the promising role of injectable hydrogels. However, there are a few aspects that could be further addressed or clarified:

Thank you very much for your constructive comments on our manuscript. The manuscript was carefully revised according to your comments. The point-by-point responses were listed below.

(1) The introduction is informative but quite dense. Consider breaking down complex sentences into simpler ones to enhance readability. Provide clear transitions between different sections, ensuring a smooth flow from the introduction to the specific focus on injectable hydrogels.

Thank you for the critical question. Following the reviewer’s suggestion, we modified the sentences in the revised manuscript to improve readability.

(2) While the introduction provides a broad context, it would be beneficial to introduce the specific types of injectable hydrogels earlier and elaborate on their unique properties. 

Thank you for the critical question. Following the reviewer’s suggestion, we have introduced specific types of injectable hydrogels early and elaborated on their unique properties in the revised manuscript.

(3) It would be valuable to include a dedicated section on smart hydrogels in the manuscript, discussing their unique characteristics and applications. Additionally, citing relevant works, such as the studies (https://doi.org/10.3390/ijms23073665) and (https://doi.org/10.1021/acsbiomaterials.0c00988), would strengthen the discussion on recent advancements in smart hydrogel research and their potential contributions to the field of injectable hydrogels for tissue filling.

Thank you for the critical question. Smart hydrogels, based on stimuli-responsive polymers with sol-gel conversion capabilities, can response to external physicochemical parameters such as temperature or pH value to form hydrogels. We have described and discussed it in the form of scattering across different parts of the manuscript, such as in section 2.1.3, 2.2.2 and 5.4, rather than a single section. Besides, we have added descriptions about the suggested references in the revised manuscript.

(4) The manuscript mentions "9 unique properties" but doesn't enumerate or explain them in the abstract. Consider providing a concise list or brief elaboration for clarity.

Thank you for the critical question. We have enumerated the unique properties of injectable hydrogel for tissue filling in the revised manuscript.

(5) The manuscript mentions intrinsic and extrinsic causes of tissue aging. It would be valuable to include a brief discussion on how injectable hydrogels specifically target or address these causes at a cellular or molecular level.

Thank you for the critical question. Since the second question above suggests a brief background in the introductory section, we have removed the intrinsic and extrinsic causes of tissue aging section from the revised manuscript.

(6) Emphasize the advantages and disadvantages of different types of injectable hydrogels, as mentioned in the latter part of the introduction, to give readers a clearer understanding of the trade-offs involved in their use.

Thank you for the critical question. Following the reviewer’s suggestion, we have added a discussion of the advantages, and disadvantages of different types of injectable hydrogels in the synthesis mechanism section of the revised manuscript.

(7) Elaborate more on the mechanisms of action and classification of injectable hydrogels. For example, discuss how different types of cross-linking affect the stability, mechanical properties, and biocompatibility of hydrogels.

Thank you for the critical question. Following the reviewer’s suggestion, we elaborate on the synthesis mechanism of injectable hydrogels in the revised manuscript and discuss how different types of crosslinking affect the stability, mechanical properties, and biocompatibility of hydrogels.

(8) Expand on the biological functions of injectable hydrogels and their therapeutic potential. How do these hydrogels interact with host cells, and what role do they play in tissue regeneration beyond mechanical support?

Thank you for the critical question. Following the reviewer’s suggestion, we have expanded the biological function of injectable hydrogels and their therapeutic potential in the revised manuscript. In section 4.3, the influences of hydrogel on host cells and tissue regeneration were added.

(9) The abstract mentions recent advancements but does not delve into specific recent studies or breakthroughs. Include more recent examples to highlight the current state of research.

Thank you for the critical question. Recent research advances are mentioned in the abstract, and in our manuscript, in section 5, specific studies or breakthroughs in injectable hydrogels are discussed in depth. It also includes some more recent examples to highlight the current state of research.

(10) Explicitly mention some of the prevailing challenges in the field to give readers a realistic view of the limitations and areas that need further exploration.

Thank you for the critical question. Following the reviewer’s suggestion, we have made detailed revisions to the outlook section to give readers a realistic view of the limitations and areas that need to be further explored.

(11) Consider adding figures or illustrations to aid in understanding the concepts discussed, especially in the section about dynamic covalent chemical bonding.

Thank you for the critical question. In the section of dynamic covalent bonding, we have drawn Figure 1 to illustrate the principle of injectable hydrogels based on dynamic covalent chemical bonding to help understand the concepts discussed. If we draw the corresponding chemical dynamic bond diagram, we will repeat the concept illustrated in Figure 1.

Minor editing of the English language is required!

Thank you for the critical question. We have edited the language in the revised manuscript.

Reviewer 4 Report

Comments and Suggestions for Authors

Summary 

The manuscript entitled "Synthesis and properties of injectable hydrogel for tissue filling" by Xie et al. is directed toward reviewing the current extensive know-how on the development of injectable hydrogels for biomedical application. The authors showed and clarified different issues related to the revised systems, and several types of approaches to using them have been discussed.

General comments 

In general, the work is accurate and clearly presented, and the given outcomes are of interest to the readers of the Pharmaceutics. 

The work contains an overview of the presented material applications, which is a timely topic and of interest to Pharmaceutics readers, as the scientific community is strongly using  injectable hydrogel materials for these applications. A full understanding of the use of these nanostructures is lacking, and this review attempts to address this topic.       

Anyway,  I believe that the text needs some technical adjustments to be published. Therefore, I recommend that this manuscript be published in the Pharmaceutics after Minor Revision. 

Going into details on the specific issues, here are some comments reported:

- The authors refer quite often (even in the Title) to ’’tissue filling’’  as a fine application. Iwould like to point out that this is not the final pourpose. A tissue cavity is not filled just because you want to fill it, but the final aims are other (e.g., tissue regeneration, healing, etc)

- One of the section is entitle ‘’’ Mechanisms of injectable hydrogels’’. I should point out that the word ‘’mechanism’’ should not refer to an object, but to an action. Therefor it should be modified, for example adding the word ‘’’shyntesis’’ at the end.

- 3. Composition of injectable hydrogels. A subsection dedicated to the description of hydrogel composites is necessary. This branch became dominant nowdays (e.g., [https://doi.org/10.1039/D3TB02693K]) and it cannot be skipped. Please add it and cite the mentioned article.  

- An entire section dedicated solely to Future Perspectives/Outlooks is necessary

Conclusion

The topic of this manuscript falls within the scope of the Pharmaceutics. I like the structure and content reported in this paper; moreover, the manuscript includes an in-depth discussion on the chemical, structural, and applicative features of the discussed materials. Anyway, I think the manuscript needs a few improvements. I believe the article is of sufficient quality to meet the Pharmaceutics publication standards after a Minor Revision. 

Comments on the Quality of English Language

- The article's English grammar and style are not correct; therefore, it should be deeply reviewed.

Author Response

Summary 
The manuscript entitled "Synthesis and properties of injectable hydrogel for tissue filling" by Xie et al. is directed toward reviewing the current extensive know-how on the development of injectable hydrogels for biomedical application. The authors showed and clarified different issues related to the revised systems, and several types of approaches to using them have been discussed.
General comments 
In general, the work is accurate and clearly presented, and the given outcomes are of interest to the readers of the Pharmaceutics. 

The work contains an overview of the presented material applications, which is a timely topic and of interest to Pharmaceutics readers, as the scientific community is strongly using injectable hydrogel materials for these applications. A full understanding of the use of these nanostructures is lacking, and this review attempts to address this topic.       

Anyway, I believe that the text needs some technical adjustments to be published. Therefore, I recommend that this manuscript be published in the Pharmaceutics after Minor Revision. 

Going into details on the specific issues, here are some comments reported:

Thank you very much for your high evaluation and constructive comments on our manuscript. The manuscript was carefully revised according to your comments. The point-by-point responses were listed below.

  1. The authors refer quite often (even in the Title) to ’’tissue filling’’  as a fine application. I would like to point out that this is not the final pourpose. A tissue cavity is not filled just because you want to fill it, but the final aims are other (e.g., tissue regeneration, healing, etc)

Thank you for the critical question. Tissue filling includes degradable filling and non-degradable filling. Degradable tissue fillers are non-permanent and will be metabolized by the body. They could stimulate cell proliferation, collagen fibers and blood vessel regeneration to regenerate tissue at the filling site, resulting in a long-lasting filling effect. Non-degradable tissue filling is permanent, does not stimulate tissue regeneration, and will not be metabolized by the body to achieve a long-lasting filling effect. In the manuscript, we mainly discuss the slow degradation rate of biodegradable injectable fillers, which have biological functions, such as promoting cell proliferation, collagen deposition and angiogenesis, to facilitate the regeneration of tissues at the filling site and achieve the effect of tissue filling.

  1. One of the section is entitle ‘’’Mechanisms of injectable hydrogels’’. I should point out that the word ‘’mechanism’’ should not refer to an object, but to an action. Therefor it should be modified, for example adding the word ‘’’shyntesis’’ at the end.

Thank you for the critical question. Following the reviewer’s suggestion, we have revised the title of Section 2 to "Synthesis mechanism of injectable hydrogels" in the revised manuscript.

  1. Composition of injectable hydrogels. A subsection dedicated to the description of hydrogel composites is necessary. This branch became dominant nowdays (e.g., [https://doi.org/10.1039/D3TB02693K]) and it cannot be skipped. Please add it and cite the mentioned article.

Thank you for the critical question. In Section 3, natural and synthetic polymers are listed, which not only include the preparation of hydrogels from only one polymer, but also the preparation of composite polymer hydrogels from the mixing of multiple polymers or inorganic hybrid polymers. Besides, we have cited the mentioned reference in the revised manuscript.

  1. An entire section dedicated solely to Future Perspectives/Outlooks is necessary.

Thank you for the critical question. Following the reviewer’s suggestion, we have made detailed revisions to the outlook section to give readers a realistic view of the limitations and areas that need to be further explored.

Conclusion

The topic of this manuscript falls within the scope of the Pharmaceutics. I like the structure and content reported in this paper; moreover, the manuscript includes an in-depth discussion on the chemical, structural, and applicative features of the discussed materials. Anyway, I think the manuscript needs a few improvements. I believe the article is of sufficient quality to meet the Pharmaceutics publication standards after a Minor Revision. 

The article's English grammar and style are not correct; therefore, it should be deeply reviewed.

Thank you for the critical question. We have edited the language in the revised manuscript.

Round 2

Reviewer 3 Report

Comments and Suggestions for Authors

All comments have been addressed well and the revised manuscript can be accepted for publication.